# Study on the mechanical properties of unloaded damage sandstone under dry-wet and freeze-thaw cycling conditions

**Wenrui Wang, Xingzhou Chen**[ID]*, **Lili Chen, Sheng Gong, Zhenkun Su**

School of Architecture and Civil Engineering, Xi'an University of Science and Technology, Xi'an, China

* xzchen0416@xust.edu.cn

**Data Availability Statement:** All relevant data are within the paper and its Supporting Information files. We have ensured that all relevant data has been included in the paper. All data in the

## Abstract

Understanding he impact of dry-wet and freeze-thaw cycles on the mechanical properties of unloaded damaged rock masses in reservoir bank slopes is crucial for revealing the deformation and failure mechanisms in artificially excavated slope rock masses within fluctuation zones. To address, the study focuses on unloaded damaged samples subjected to excavation disturbances, conducting various cycles of dry-wet and freeze-thaw treatment along with uniaxial and triaxial re-loading tests. A damage statistical constitutive model was established based on the experimental results and validated using numerical simulation methods. The results indicate: (1) The mechanical properties of sandstone, which has incurred damage and is not under load, are significantly impacted by cycles of drying-wetting and freezing-thawing. As the number of these environmental cycles increases, a descending trend becomes apparent in the stress-strain curve profile. This shift coincides with an increase in pore compaction strain as well as plastic strain values; meanwhile, peak strength experiences a sharp decline initially but subsequently moderates to more gradual reductions.; (2) The elastic modulus, cohesion, and friction angle all show a similar trend of attenuation, with the most severe degradation occurring after the first cycle and then gradually diminishing, particularly with the elastic modulus; (3) The uniaxial failure of the unloaded damaged samples is primarily brittle, with spalling and buckling becoming more pronounced with increasing cycles, while triaxial failure exhibits certain plastic characteristics that develop more with further cycling. As the frequency of dry-wet and freeze-thaw cycles rises, there is a corresponding increase in the number of fractures observed at the point of sample failure.

## Introduction

Reservoir bank slope stability is crucial for the construction and safe operation of reservoirs. Various factors, including artificial excavation disturbances, periodic water level adjustments, and seasonal variations, have a direct impact on the stability of slopes along reservoir banks [1–3]. The slopes of these rock masses are subjected to a range of stress factors. These include the damage caused by unloading during engineering excavations [4], the deterioration effects of dry-wet cycling due to water level variations following reservoir construction [5], and

manuscript are the property of Prof Chen Xingzhou's group at the School of Architecture and Civil Engineering, Xi'an University of Science and Technology.

**Funding:** This research was funded by THE NATIONAL NATURAL SCIENCE FOUNDATION OF CHINA, grant number U1965107.The funders had no role in study design, data collection and analysis, decision to publish, or preparation of the manuscript.

**Competing interests:** The authors have declared that no competing interests exist.

erosion impacts from the freeze-thaw processes observed with seasonal changes [6]. Such a combination of factors adds complexity to the evaluation of mechanical characteristics in rock masses within the fluctuation zone of artificially cut bank slopes. Therefore, investigating the mechanical properties of sandstone, particularly when impact by unloading damage in conjunction with dry-wet and freeze-thaw cycling conditions, is of significant engineering importance.

For extensive slope excavation initiatives, the methodology typically entails a phased approach involving both excavation and unloading. The impact of these processes on the mechanical characteristics of rock masses has been the subject of thorough investigations by researchers such as Qiu et al. [7], Liang et al. [8], Zhu et al. [9], and Zhang et al. [10]. These studies, utilizing a blend of laboratory experimentation, practical engineering applications, and numerical simulation techniques, have not only contributed to a deeper understanding of the mechanical dynamics in unloaded rock masses but have also highlighted the significant impact of unloading damage, a byproduct of excavation, on the mechanical properties of rocks constituting slopes. The rock structure deterioration under dry-wet cycling represents a macroscopic manifestation of damage to mechanical properties. Liu et al. [11], Yuan et al. [12], Chen et al. [13], Sun et al. [14], and Zhang et al. [15] conducted studies on the degradation of macroscopic strength indicators and microscopic pore changes. To accurately describe the deformation characteristics under dry-wet cycling, several scholars have used statistics and damage mechanics to describe the constitutive relationship of the basic laws of rock deformation strength. Huang et al. [16, 17] and Hu et al. [18] have established damage degradation constitutive models describing the stress-strain relationship under different dry-wet cycling conditions based on statistical damage mechanics. In cold climates with changing seasons, rock bodies inevitably undergo freeze-thaw actions. The phase change of water to ice in rock bodies causes changes in the internal granular structure of the rock due to frost heave forces. Momeni et al. [19], Ma et al. [20] investigated the degradation of rock mechanical properties under freeze-thaw cycling using laboratory experiments and numerical simulation methods. Further to these observations, many researches documented a heightened vulnerability of rock materials' mechanical traits when subjected to repeated dry-wet and freeze-thaw cycles. Özbek [21] and Sun et al. [22] delved into the attrition of rocks at both macroscopic mechanical levels and microscopic structural dimensions under cycling environmental conditions akin to drying-wetting and freezing-thawing processes. Despite this interest in degradation patterns, there remains an evident dearth of controlled indoor experiments rigorously addressing dry-wet and freeze-thaw sequences [23]; Specifically, detailed protocols for accurately emulating these cyclic events are missing. To capture the complexity involved in depicting laws governing mechanical property decay, particularly within complex scenarios involving varying climatic factors common along slopes adjoining reservoirs [24]–It is vital that simulations correctly reflect natural sequels exerted by different contributing elements inherent to those ecosystems.

To obtain the mechanical properties of the unloaded damaged rock body under the condition of dry-wet-freeze-thaw cycle and the change rule of the mechanical properties under different times of cyclic action, it is proposed that the stress-strain response mechanism of the unloaded rock body can be scientifically responded to the different times of dry-wet-freeze-thaw cyclic action. Accordingly, the research focused on unloaded damaged samples crafted from sandstone sourced from the vicinity of a specific reservoir's slopes. In line with water level management protocols and local meteorological conditions associated with this reservoir region, comprehensive dry-wet and freeze-thaw cycle tests were orchestrated for these specimens. Post cyclic conditioning, uniaxial as well as triaxial re-loading assessments were performed. Leveraging statistical damage mechanics principles, a compression-induced

deterioration model was formulated accounting. The research aims to provide a theoretical reference for disaster prevention and mitigation at the fluctuating zone of artificially excavated slopes within reservoirs. Additionally, it seeks to promote the development of theoretical systems for analyzing the mechanical properties of unloaded rock formations under diverse conditions.

## Sample preparation and test methods

### Sample selection and preparation

Rock samples for the experiment were collected from red sandstone in the engineering area of a reservoir. In compliance with International Standard for Results Management (ISRM) standards, the specimens were shaped into cylinders, uniform in design, measuring 50 mm in diameter and 100 mm in height. This standardization process was employed to minimize sample variation. Concurrently, acknowledging that the rock structures within the reservoir's drawdown zone had previously been subjected to excavation-based unloading stressors, laboratory-induced simulations mirroring these conditions were carried out on pristine materials to produce representative samples exhibiting characteristics typically resulting from such disturbances. The preparation involved specific procedural steps aimed at replicating those damages:

A triaxial compression test (confining pressure = 9 MPa) was conducted to obtain a triaxial peak compressive strength as $\sigma_{SC}$ = 115 MPa.

A triaxial unloading test was employed at the axial stress level 70% of $\sigma_{SC}$ to obtain the confining pressure value of 2.5 MPa at the time of unloading failure.

The extent of unloading damage was quantified using the amount of unloading [25]. The unloading quantity is defined as follows:

$$U_s = \frac{\sigma_3^0 - \sigma_3^i}{\sigma_3^0 - \sigma_3^f} \times 100\% \tag{1}$$

Where, $U_s$ represents the amount of unloading; $\sigma_3^0$ represents the initial perimeter pressure value before unloading; $\sigma_3^i$ represents the target perimeter pressure value for unloading; and $\sigma_3^f$ represents the perimeter pressure value at the time of unloading damage.

To ensure the typicality of unloading damage and the smooth execution of subsequent dry-wet and freeze-thaw cycling tests, a 60% unloading (resulting in a confining pressure of 5.1 MPa) was selected for preparing the unloading damaged samples. This decision was made with consideration for the practical applicability of the engineering rock body.

Fig 1 illustrates the stress path utilized in preparing the sample with unloading damage for the test. Concurrently. Fig 2 presents the characteristic stress-strain curve observed during the testing procedure.

### Cycling test methods

Based on the water level regulation needs and climatic environmental factors of the reservoir area, a dry-wet and freeze-thaw cycling path was established. The reservoir undergoes multiple water level adjustments throughout the year, and the area experiences conditions suitable for freeze-thaw cycles during the spring and winter seasons due to temperature changes. Therefore, the dry-wet and freeze-thaw cycling path was comprehensively set to include one freeze-thaw cycle after every three dry-wet cycles (Fig 3).

To as accurately as possible replicate the environmental conditions of the unloaded damaged rock body in the drawdown zone while also shortening the experimental duration, the

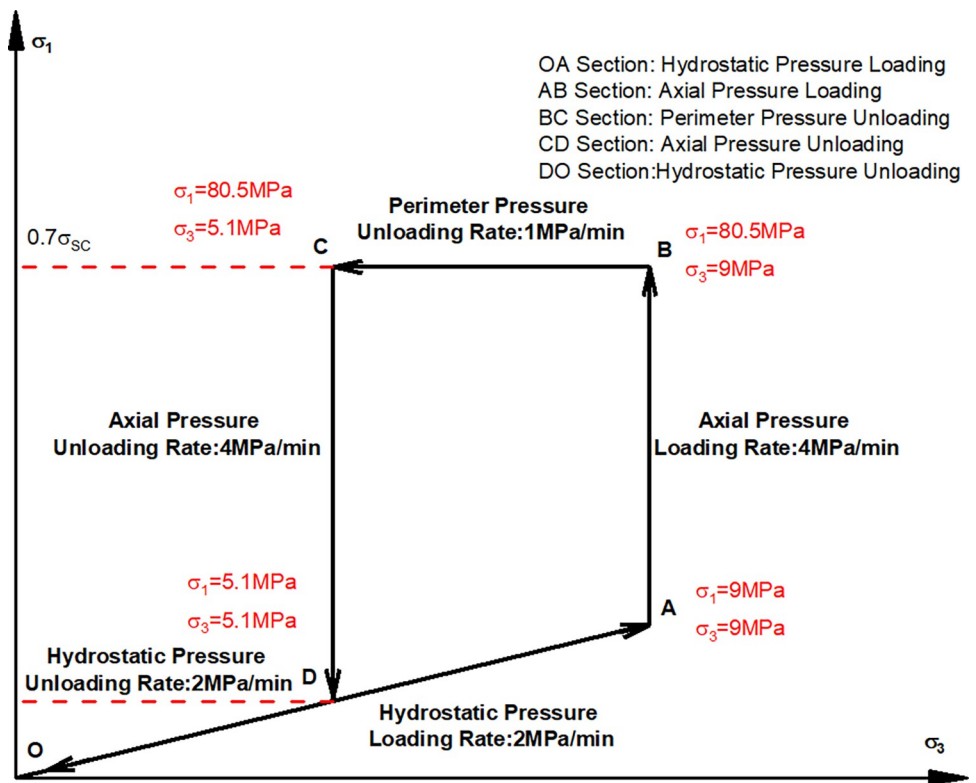

**Fig 1. Stress path for the preparation of unloaded damaged sample.**

drying temperature and time were chosen as 105°C for 24 h. The saturation method and time involved placing the sample in a pressurized soaking saturation tank, adding water to ensure the water level was above the sample, pressurizing until the pressure gauge read 0.3 MPa, and soaking for 24 h to ensure full saturation of the unloaded damaged sample. For freezing methods, temperature and time were chosen considering the in-situ environment of the unloaded damaged rock body and the triaxial constraint effects that prevent frost heave force from reducing through material deformation. To simulate a real freeze-thaw state, a homemade rock specimen freezing test fixture was used. The sandstone specimen was secured within a mounting apparatus, ensuring that during the setup process, bolts were fastened to induce an initial pre-tension force. Immediately after this preparatory step, without delay it was transferred into a refrigeration unit capable of low-temperature conditions. The temperature inside this freezer facilitator was meticulously calibrated at -25°C for the sample's exposure duration which spanned 12 h.

Post-completion of 1st, 3rd, 5th, 7th, and ultimately up to the 9th cycle iteration entailing dry-wet as well as freeze-thaw treatments—each unloaded damaged specimen underwent extraction for mechanical testing. The suite of tests executed included both uniaxial compression examination along with triaxial re-loading procedures.

The purpose of the post-cyclic reloading test on unloaded damaged specimens is to obtain the me-chamisal properties of the unloaded damaged specimens after different numbers of cyclic actions. Among them, uniaxial reloading is to place the unloaded damaged specimen that has been cycled to a certain num-ber of times in the rock uniaxial loading apparatus and load the damage at a rate of 4MPa/min. It should be noted that during the triaxial reloading test, the perimeter pressure value was set to the target value of perimeter pressure unloading at

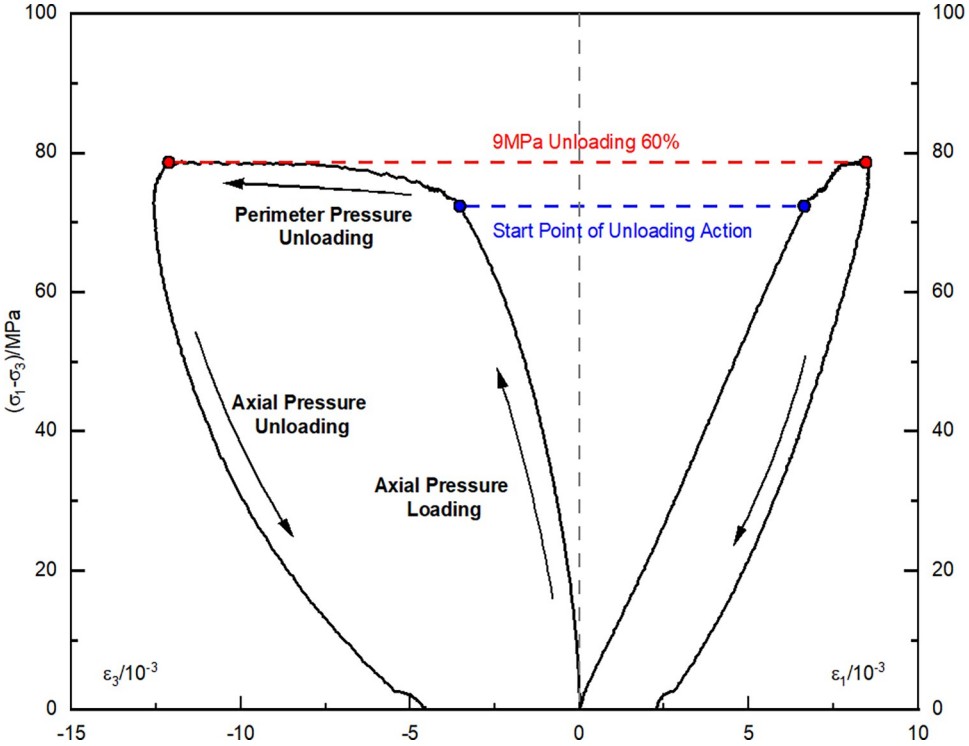

**Fig 2. Stress-strain curve for the preparation of unloaded damaged sample.**

the time of unloading damage rock sample preparation. After placing the specimen in the tri-axial reloading test apparatus, the axial pressure and peripheral pressure are synchronously loaded to the target value (5.1MPa) of peripheral pressure unloading at a rate of 2 MPa/min to restore the original stress state of the specimen before the cyclic test, and then the axial pressure is loaded at a rate of 4 MPa/min until the specimen is damaged. During the test, the stress and strain values during loading and at the time of destruction were collected by means of an axial displacement strain gauge.

## Test results

### Stress-strain curve analysis

Using the unloaded damaged samples that had not undergone any cycling as the initial state, uniaxial and triaxial re-loading tests were conducted. The resulting stress-strain curves were depicted in Figs 4 and 5.

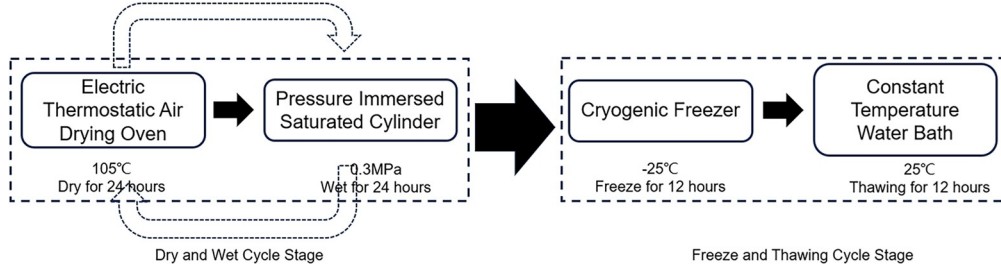

**Fig 3. Schematic diagram of dry-wet and freeze-thaw cycling path.**

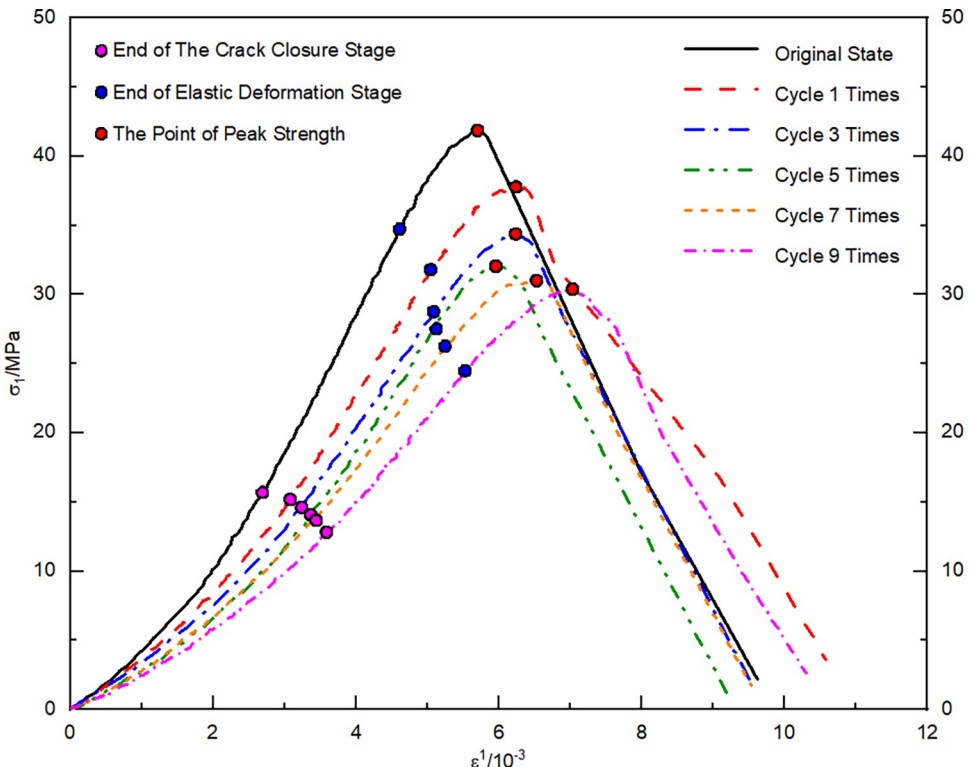

**Fig 4. Uniaxial re-loading stress-strain curve after dry-wet and freeze-thaw cycling.**

The morphological analysis of stress-strain curves from unloaded damaged samples, subjected to a variety of loading conditions and subjected to numerous dry-wet and freeze-thaw cycles, consistently exhibits four primary stages: compaction of cracks, elastic behavior, plastic deformation, and the stage subsequent to structural failure. In the crack compaction phase, the curve initially shows a concave profile, indicative of initial structural adjustments within the sample. With increasing environmental cycle exposure, the pronounced nature of the concavity noted in the stress-strain curve is mitigated by confining pressure. This data suggests that cyclic exposure to drying-wetting and freezing-thawing processes promotes the development and enlargement of micro-pores and cracks within the material matrix. Conversely, confining pressure acts to compress these micro-voids, thereby diminishing their prominence. As the material transitions into the elastic phase, a consistent linear relationship between stress and strain is observed; however, the slope of this relationship gradually lessens as the cycle count rises. This change is indicative of the material's progressive propensity for deformation when subjected to load stress, in tandem with a reduction in its elastic modulus. The plastic phase is characterized by a transition of the curve into a convex form, with a noticeable acceleration in the rate of strain growth. As the cycle count increases, the extent of the plastic region in the curve extends, signifying a progressive 'softening' of the material's properties. The final phase, post-failure, is marked by a rapid decline in the stress-strain curve following its peak. This decline signifies the point where the deviatoric stress surpasses the material's shear capacity, leading to a swift reduction in the load-bearing ability of the sample as it undergoes further deformation.

Statistical analysis was conducted on the stress and strain values at key points in the stress-strain curve, including the compaction point, the onset of plastic yield, and peak stress point, (Table 1).

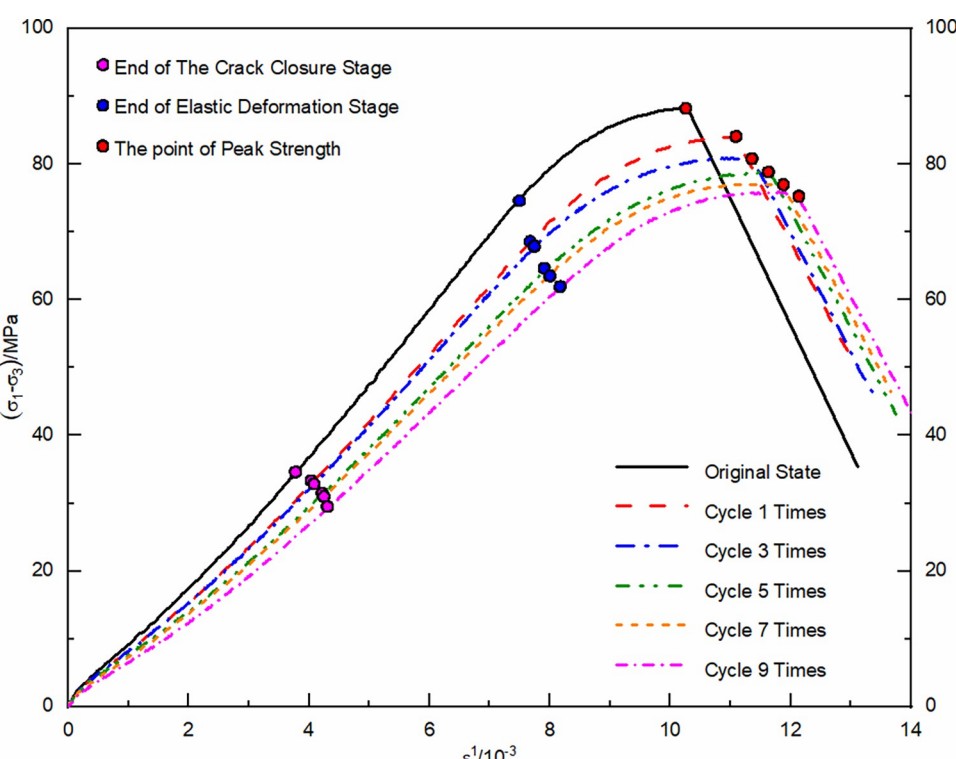

**Fig 5. Triaxial re-loading stress-strain curve after dry-wet and freeze-thaw cycling.**

With each successive dry-wet and freeze-thaw event, stress at compaction points tends to reduce, and strain values rise accordingly. Elevated confining pressures yield higher stress at compaction points, with negligible increases in strain. This phenomenon originates from the development and expansion of pores and micro-cracks within the unladen sample, a result of the repeated environmental cycles. Consequently, this leads to a reduction in the deviatoric stress required for pore compaction, while the strain resulting from pore closure gradually intensifies. Within the realm of confining pressure, initial hydrostatic pressure stages result in

**Table 1. Summary of key stress-strain points in re-loaded unloaded damaged samples under dry-wet and freeze-thaw cycling conditions.**

| Number of Cycles/n | Confining Pressure/MPa | Compaction Point | | Beginning of Plastic Yield | | Peak Point | |
|---|---|---|---|---|---|---|---|
| | | Deviatoric Stress/MPa | Strain/$10^{-3}$ | Deviatoric Stress/MPa | Strain/$10^{-3}$ | Deviatoric Stress/MPa | Strain/$10^{-3}$ |
| Initial State | 0 | 15.972 | 2.735 | 34.558 | 4.576 | 41.956 | 5.722 |
| | 5.1 | 26.966 | 3.045 | 74.650 | 7.485 | 88.417 | 10.274 |
| 1 | 0 | 14.486 | 2.997 | 31.922 | 5.059 | 37.764 | 6.369 |
| | 5.1 | 26.446 | 3.333 | 68.657 | 7.676 | 84.010 | 11.090 |
| 3 | 0 | 13.966 | 3.003 | 28.498 | 5.071 | 34.325 | 6.290 |
| | 5.1 | 26.375 | 3.358 | 67.797 | 7.741 | 80.595 | 11.302 |
| 5 | 0 | 13.949 | 3.507 | 27.596 | 5.098 | 32.079 | 5.986 |
| | 5.1 | 26.244 | 3.597 | 64.366 | 7.930 | 78.839 | 11.537 |
| 7 | 0 | 13.355 | 3.528 | 26.005 | 5.234 | 31.075 | 6.560 |
| | 5.1 | 26.531 | 3.716 | 63.548 | 7.991 | 76.980 | 11.873 |
| 9 | 0 | 13.033 | 3.651 | 24.334 | 5.516 | 30.423 | 6.963 |
| | 5.1 | 25.100 | 3.784 | 61.876 | 8.176 | 75.828 | 11.839 |

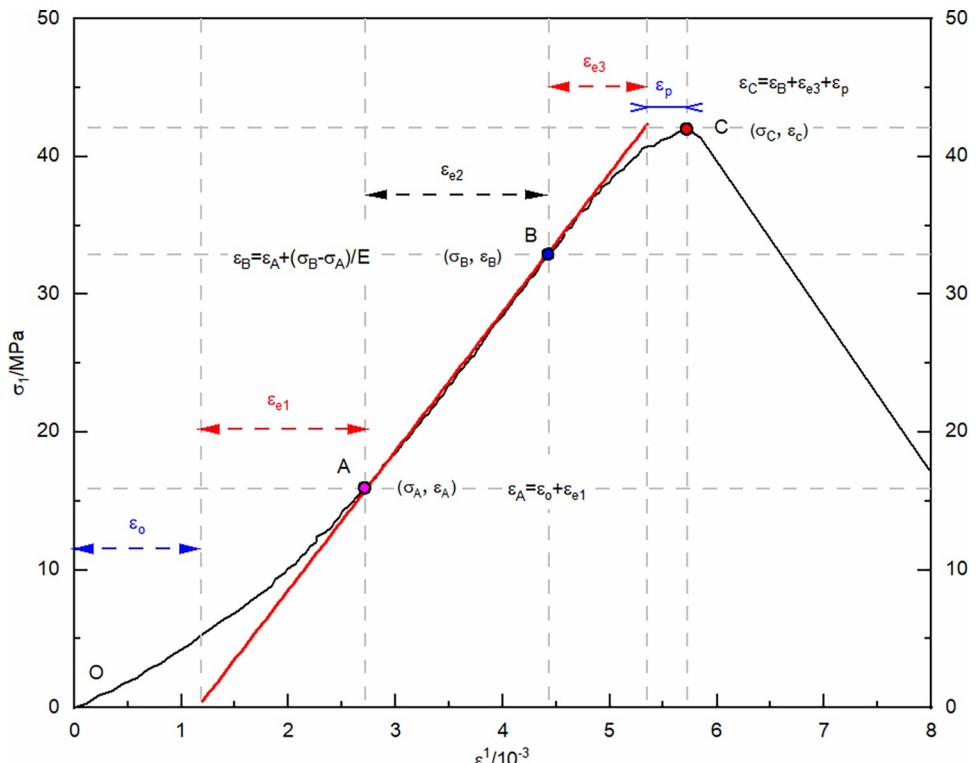

**Fig 6. Schematic diagram of the stress-strain relationship during the re-loading process.**

the compression of pores. This leads to a more gradual increase in the strain value at the compaction point, especially when compared to the uniaxial strain value, for escalating numbers of dry-wet and freeze-thaw cycles. As the cycle number grows, the start of plastic yield continuously shifts downwards, shortening the elastic stage and leading the sample to enter the plastic deformation state more quickly. The peak stress values show a nonlinear decreasing trend with an increasing number of cycles, while peak strains slightly increase.

Strain composition analysis was conducted based on the stress-strain curve of uniaxial re-loading without previous cycling, (Fig 6). Elastic deformation permeates the entire compression process in the unloaded damaged samples. During the initial compaction stage, deformation occurs due to pore closure. As axial loading continues, the deformation caused by pore closure gradually decreases, and the unloaded damaged samples are increasingly compacted. During the plastic yield stage, the sample undergoes plastic deformation, and as axial loading continues, the number of particles in the unloaded damaged samples undergoing plastic deformation gradually increases, and the plastic deformation increases as well. Statistical analysis of the strain composition allows for the analysis and forecasting of the strain development pattern under load for unloaded damaged samples with varying cycling numbers.

The compaction point stress and strain are represented by $(\sigma_A, \varepsilon_A)$, where its strain is the sum of pore compaction strain, $\varepsilon_0$, and the elastic strain, $\varepsilon_{e1}$, accrued during compaction. Plastic yielding onset stress and strain are denoted by $(\sigma_B, \varepsilon_B)$, incorporating strain from both the compaction phase and the elastic deformation's elastic strain, $\varepsilon_{e2}$. At the peak stress point, referred to as $(\sigma_C, \varepsilon_C)$, strain encompasses contributions from compaction, elasticity, and plasticity, namely elastic strain $\varepsilon_{e3}$ and plastic strain $\varepsilon_p$. To elucidate the impact of dry-wet and freeze-thaw cycle frequency on the strain-stress profiles of sandstone unaffected by load, curve fitting for both compaction point stress-strain and pore compaction strain, as well as peak

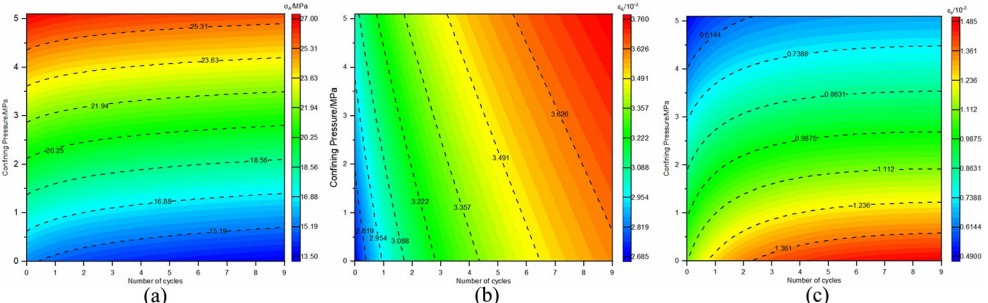

**Fig 7. Relationship between stress-strain at compaction point and the number of dry-wet and freeze-thaw cycle, with re-loading confining pressure.**

point stress-strain and plastic strain under various confining pressures, is executed. The established correlations between compaction and peak point stress-strain with the number of dry-wet and freeze-thaw cycles, as well as re-loading confining pressures, are illustrated in Figs 7 and 8.

As the frequency of dry-wet and freeze-thaw cycles escalates, there is a consistent reduction in stress values at the compaction point, while strain values exhibit a corresponding rise, leading to an increment in pore compaction strain. As the confining pressure value of re-loading increases, the stress values at the compaction point continuously increase, as do the strain values, while the pore compaction strain decreases. The underlying cause is attributed to the repetitive dry-wet and freeze-thaw cycles that progressively augment the pores and cracks in the unloaded, damaged samples, thereby gradually loosening the structure. With each successive dry-wet and freeze-thaw cycle, there is a consistent decrease in the stress values at the peak point, accompanied by a corresponding increase in the strain values. Under lower confining pressures, the plastic strain remains relatively unchanged, whereas at higher confining pressures, there is a noticeable increment in the plastic strain. As the re-loading confining pressure value increases, the stress, strain, and plastic strain values continuously increase. The reason for this is that at lower confining pressures, the samples primarily exhibit brittle failure characteristics, but as the confining pressure increases, the unloaded damaged samples exhibit plastic failure characteristics. The enhancement of plastic characteristics in samples that have not been subjected to loading is observed as a consequence of an elevated number of dry-wet and freeze-thaw cycles.

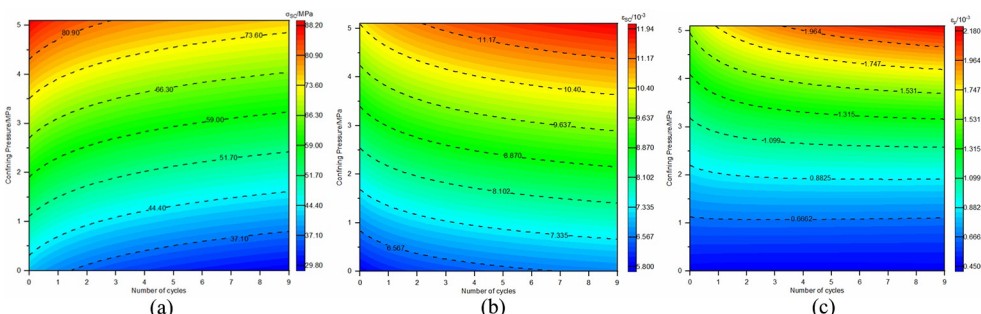

**Fig 8. Relationship between stress-strain at peak point and the number of dry-wet and freeze-thaw cycle, with re-loading confining pressure.**

**Table 2. Summary of changes in deformation strength parameters of unloaded damaged samples upon re-loading to the number of the dry-wet and freeze-thaw cycle.**

| Number of Cycles/n | Elastic Modulus/MPa | | Cohesion/MPa | Angle of Friction/° |
|---|---|---|---|---|
| | Uniaxial Re-loading | Triaxial Re-loading | | |
| Initial State | 10093.798 | 10683.552 | 6.555 | 55.178 |
| 1 | 8253.504 | 9640.853 | 5.931 | 55.062 |
| 3 | 7458.066 | 9346.349 | 5.431 | 54.962 |
| 5 | 7793.449 | 8761.281 | 5.144 | 54.883 |
| 7 | 6848.909 | 8660.721 | 4.953 | 54.805 |
| 9 | 6006.012 | 8310.816 | 4.834 | 54.742 |

## Mechanical parameter analysis

The deformation strength parameter results of unloaded damaged samples with varying numbers of dry-wet and freeze-thaw cycle numbers are presented in Table 2.

An inverse relationship exists between the elastic modulus and the frequency of dry-wet and freeze-thaw cycles, whereas a direct relationship is found between the elastic modulus and confining pressure. An increase in the cycle frequency leads to a gradual decline in the elastic modulus, but the presence of higher confining pressures tends to offset this reduction, leading to an enhanced elastic modulus. Furthermore, higher confining pressures mitigate the degree of elastic modulus deterioration due to increased cycling. Therefore, curve fitting for the elastic modulus was conducted with respect to the number of dry-wet and freeze-thaw cycles as well as various confining pressure magnitudes. The result of this fitting is depicted by the surfacing Fig 9, and is mathematically expressed in Eq (2). The fitting curves for cohesion and the angle

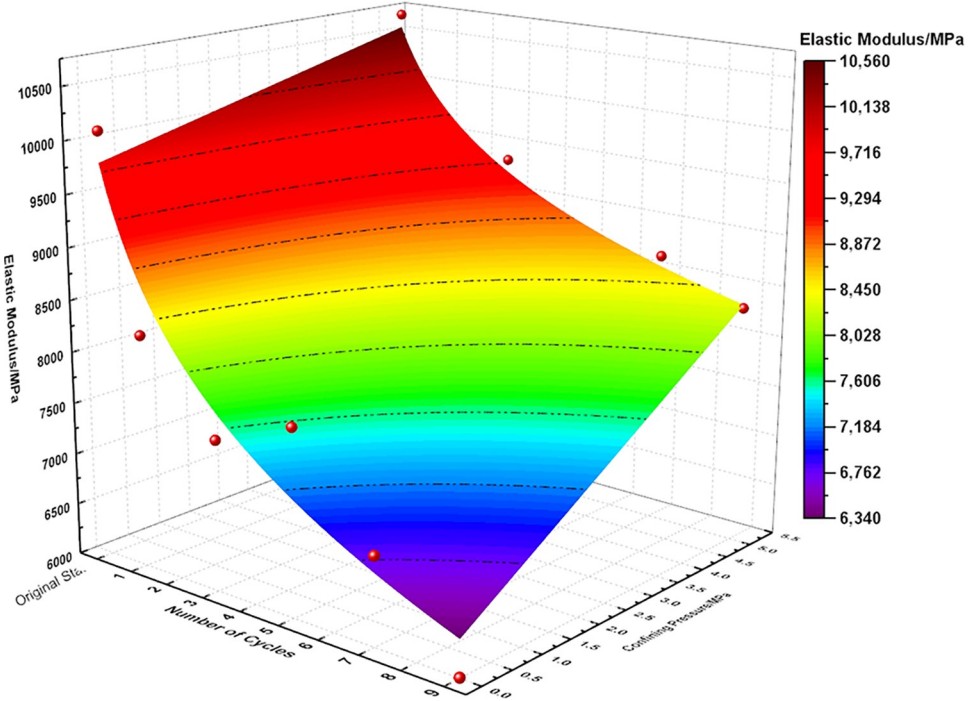

**Fig 9. Surface relationship between elastic modulus, number of dry-wet and freeze-thaw cycles, and confining pressure.**

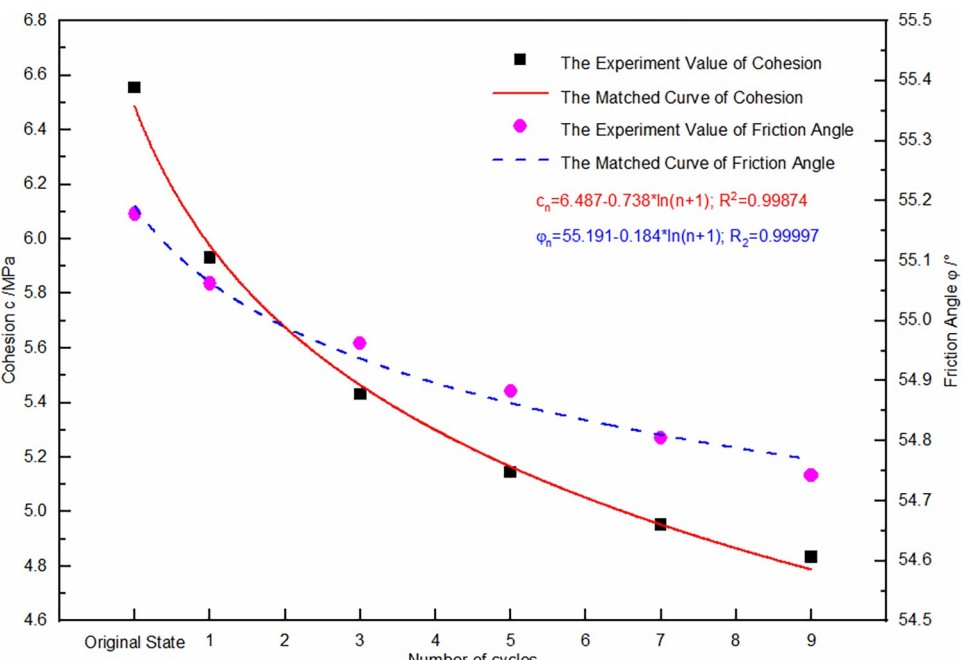

**Fig 10. Relationship curve between cohesion and friction angle with the number of dry-wet and freeze-thaw cycles.**

of friction are also shown in Fig 10, with their respective expressions in Eqs (3) and (4).

$$E_n = (148.308 \cdot \sigma_3 + 9796.621) + (104.810 \cdot \sigma_3 - 1493.475) \cdot ln(n+1);$$

$$R^2 = 0.92251 \tag{2}$$

$$Cn = 6.487 - 0.738 \cdot ln(n+1); R^2 = 0.99874 \tag{3}$$

$$\varphi_n = 55.191 - 0.184 \cdot ln(n+1); R^2 = 0.98997 \tag{4}$$

Fig 9 illustrates that the elastic modulus demonstrates a non-linear decline in deterioration under conditions of dry-wet and freeze-thaw cycling, with an initial steep decrease followed by a more gradual trend. The first cycle shows an approximate 10% to 18% degradation in elastic modulus, and a power function related to confining pressure fits well, indicating that an increase in confining pressure slows down the degradation of the elastic modulus. Cohesion and the angle of friction also show the same degradation trend, gradually decreasing with an increasing number of cycles. The degree of cohesion degradation is quite significant, with about a 9.5% reduction after the first cycle and approximately 25% after 9 cycles. The degradation degree of the angle of friction is relatively minor, with only about a 0.8% reduction after 9 cycles.

The reason for this is that, during the wetting and drying phases of the dry-wet and freeze-thaw cycle, the lead to a reduction in strength of particles in unloaded damaged samples. This reduction occurs as sol-uble components inside the unloaded damaged sample react with water and decrease,, reducing the strength of the particles when soaked in water. During the drying process, the departure of water weakens the cementation between particles.

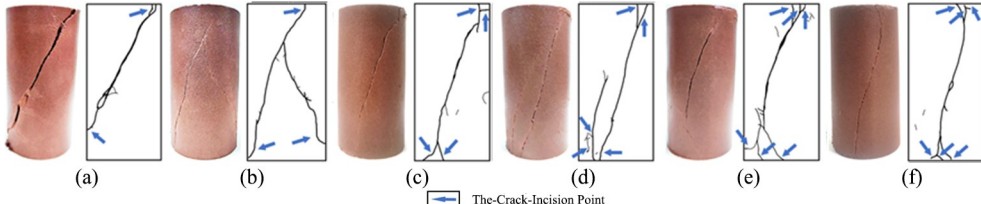

**Fig 11.** Fracture morphology of unloaded damaged samples after dry-wet and freeze-thaw cycles under uniaxial re-loading: (a) Initial sate; (b) After 1 cycle; (c) After 3 cycles; (d) After 5 cycles; (e) After 7 cycles; (f) After 9 cycles.

Throughout the freezing process, the force of frost heave, which arises from the transformation of water into ice, diminishes the cementation between particles. In the thawing process, because the expansion phenomenon caused by the phase change from ice back to water cannot fully recover, the structure of the unloaded damaged sample becomes increasingly loose. The elastic modulus, which characterizes the extent of rock material deformation under load, diminishes as the number of loading cycles increases due to the formation of internal pores and reduction in particle contact area. Consequently, this leads to less resistance to relative particle displacement. Cohesion primarily characterizes the strength of the interparticle bonding in the rock, which is disrupted due to the repeated changes in the water content of the unloaded damaged samples and the frost heave force during the dry-wet and freeze-thaw cycle. The angle of friction, which is controlled by the roughness of the particles, shows weaker changes throughout the dry-wet and freeze-thaw cycles.

## Failure characteristic analysis

The failure morphology of uniaxial re-loading is displayed in Fig 11, and the failure characteristics of triaxial re-loading are shown in Fig 12.

In the uniaxial re-loading tests of the unloaded damaged samples (Figs 11 and 12), the macroscopic failure surfaces generally exhibit a vertical longitudinal distribution with distinct tensile splitting characteristics and obvious signs of brittle failure. This type of failure is due to the generation of lateral tensile stresses in the sample, leading to the formation and progression of cracks. As the number of cycles increases, the failure mode gradually shifts to buckling and spalling, with the buckling and spalling area gradually enlarging, and larger volumes of rock blocks peeling off the sample. The more the cycles there are, the longer the persistence of buckling and spalling phenomena. In triaxial re-loading tests, the failure morphology of the unloaded damaged samples mainly presents as an oblique through-going failure plane from top to bottom. With an increasing number of cycles, the number of cracks gradually increases, especially the smaller cracks at both ends. Similarly, in triaxial re-loading tests, the unloaded

**Fig 12.** Fracture morphology of unloaded damaged samples after dry-wet and freeze-thaw cycles under triaxial re-loading (Confining pressure at 5.1 MPa): (a) Initial sate; (b) After 1 cycle; (c) After 3 cycles; (d) After 5 cycles; (e) After 7 cycles; (f) After 9 cycles.

damaged samples not previously subjected to cycles accompany a loud fracture noise at the moment of failure. This fracture noise gradually decreases with an increasing number of dry-wet and freeze-thaw cycles. Under continuous cycling, the unloaded damaged samples exhibit a tendency for brittle failure to shift toward plastic behavior.

## Damage constitutive model and numerical simulation verification

### Construction of damage model

Regarding the compressive testing of sandstone that has not been loaded and has incurred damage, the damage variable is computed and the damage constitutive model is constructed using the statistical approach delineated in Cao et al. [26]. It is postulated that the strength of microelements adheres to Weibull probability distribution, effectively characterizing the entire spectrum of rock deformation and failure [27, 28]. The probability density function is articulated as follows:

$$P(F; \lambda, k) = \frac{k}{\lambda} \left(\frac{F}{\lambda}\right)^{k-1} exp\left[-\left(\frac{F}{\lambda}\right)^k\right] \tag{5}$$

During compression, the rock's damage variable, $\omega_c$, is determined by comparing the number of failing microelements, $N_f$, within the rock to the overall count of microelements, $N$. The calculation of this damage variable, crucial for to the rock's structural stability, is impact by the probability of failure of these microelements, a factor derived from the probability density function, as elaborated below:

$$\omega_c = \frac{N_f}{N} = \frac{\int_0^F N \cdot P(x)\mathrm{d}x}{N} = 1 - exp\left[-\left(\frac{F}{\lambda}\right)^k\right] \tag{6}$$

Where: $\omega_c$ represents the damage variable during compression; $N_f$ represents the number of microelements undergoing failure inside the rock during compression; $N$ represents the total number of microelements; $F$ represents the microelement strength; while $\lambda$ and $k$ represents Weibull distribution parameters related to rock material.

The microelement strength of rock is determined using rock strength failure criteria, and Mohr-Coulomb criterion is adopted as a classical rock mechanics strength theory. It is characterized by clear physical parameters, simplicity, and efficiency in use, and can accurately reflect rock strength. It employs the principal stress representation as follows:

$$\sigma_1(1 - sin\,\varphi) - \sigma_3(1 + sin\,\varphi) = 2c\,cos\,\varphi \tag{7}$$

Drawing from the generalized Hooke's law $\sigma_2 = \sigma_3$ under biaxial conditions, which can be obtained as follows:

$$\sigma_1 = E\varepsilon_1 + 2\mu\sigma_3 \tag{8}$$

The microelement strength can be derived as follows:

$$F = \frac{E\varepsilon\left(\sigma_1 - \frac{1+sin\varphi}{1-sin\varphi}\sigma_3\right)}{\sigma_1 - 2\mu\sigma_3} \tag{9}$$

By applying the method of derivatives and considering the geometric attributes of the peak point on the stress-strain curve during loading, the model parameters were determined based on extreme value theory. This approach is utilized when the curve's slope becomes zero(i.e.,

$\partial\sigma_1/\partial\varepsilon_1 = 0$), at the point where stress attains its maximum strength. Subsequently, the pair of parameters characterizing Weibull distribution are determined as follows:

$$k = \frac{1}{ln\frac{E\varepsilon_1}{\sigma_1-2\mu\sigma_3}} \tag{10}$$

$$\lambda = \left[\frac{F^k}{ln\frac{E\varepsilon_1}{\sigma_1-2\mu\sigma_3}}\right]^{\frac{1}{k}} = F \cdot k^{\frac{1}{k}} \tag{11}$$

While considering the compaction stage, the stress-strain relationship of this stage is represented as follows:

$$\sigma_1 = E\varepsilon \cdot a\ exp(b\varepsilon) + 2\mu\sigma_3 \tag{12}$$

The proposed stress-strain relationship must satisfy: ① The stress-strain curve should pass through the compaction point B($\varepsilon_A,\sigma_A$); ② The tangent modulus at the compaction point A should be the same as the elastic modulus.

$$b = \frac{\varepsilon_o}{\varepsilon_A(\varepsilon_A - \varepsilon_o)} \tag{13}$$

$$a = \left(1 - \frac{\varepsilon_o}{\varepsilon_A}\right) \cdot exp(-b\varepsilon_A) \tag{14}$$

The identification of the compaction point and the pore compaction strain can be fitted based on the aforementioned experimental results.

$$\sigma_A = (2.258 \cdot \sigma_3 + 15.464) + (0.067 \cdot \sigma_3 - 0.851) \cdot ln(n + 1); R^2 = 0.96914 \tag{15}$$

$$\varepsilon_A = (0.00007 \cdot \sigma_3 + 0.00269) - (0.00002 \cdot \sigma_3 - 0.00040) \cdot ln(n + 1);$$

$$R^2 = 0.89725 \tag{16}$$

$$\varepsilon_0 = (\varepsilon_A \cdot E_n - \sigma_A)/E_n \tag{17}$$

Therefore, the complete damage evolution model can be represented as follows:

$$\begin{cases} \sigma_1 = E\varepsilon \cdot a\,exp(b\varepsilon) + 2\mu\sigma_3; \varepsilon \leq \varepsilon_A \\ \sigma_1 = E(\varepsilon - \varepsilon_A) \cdot exp\left[-\left(\frac{F}{\lambda}\right)^k\right] + 2\mu\sigma_3; \varepsilon > \varepsilon_A \end{cases} \tag{18}$$

Based on the experimental results previously discussed, the parameters impacting deformation strength within the damage evolution model are ascertainable. These parameters are derived from fitting expressions that incorporate the frequency of dry-wet and freeze-thaw cycles and the magnitudes of re-loading confining pressure. This methodology enables the estimation of the compressive stress-strain relationship in unloaded, damaged samples under varying confining pressures and different counts of dry-wet and freeze-thaw cycles.

**Table 3. Summary of damage evolution model parameters for unloaded damaged samples under dry-wet and freeze-thaw cycling conditions.**

| Number of Cycles/n | Confining Pressure/MPa | Compaction Segment Parameters | | Weibull Distribution Parameters | |
|---|---|---|---|---|---|
| | | $a$ | $b$ | $k$ | $\lambda$ |
| Initial State | 0 | 0.291 | 265.230 | 9.144 | 59.896 |
| | 5.1 | 0.691 | 63.867 | 6.039 | 62.941 |
| 1 | 0 | 0.268 | 249.352 | 9.920 | 52.000 |
| | 5.1 | 0.684 | 60.903 | 5.126 | 58.105 |
| 3 | 0 | 0.270 | 228.619 | 10.629 | 46.838 |
| | 5.1 | 0.691 | 56.391 | 4.725 | 54.212 |
| 5 | 0 | 0.277 | 215.066 | 11.084 | 44.206 |
| | 5.1 | 0.698 | 53.367 | 4.562 | 52.030 |
| 7 | 0 | 0.285 | 204.788 | 11.441 | 42.429 |
| | 5.1 | 0.705 | 51.061 | 4.466 | 50.488 |
| 9 | 0 | 0.292 | 196.420 | 11.742 | 41.086 |
| | 5.1 | 0.711 | 49.180 | 4.400 | 49.288 |

## Verification of damage model

By combining the fitting expressions identified for the compaction stage and the deformation strength fitting expressions, the parameters for the compaction segment of the unloaded damaged sample damage statistical evolution model under dry-wet and freeze-thaw cycling and Weibull distribution parameters are obtained, with the statistical results (Table 3).

By substituting the calculated parameter values (Table 3) into Eq (18), the theoretical curve of the damage evolution model for unloaded damaged sandstone under dry-wet and freeze-thaw cycling can be obtained. A comparison of the theoretical curve with the experimental curve is shown in Figs 13 and 14.

Under the same confining pressure conditions, a pattern emerges where the peak strength gradually reduces, whereas the peak strain exhibits an increase with an escalating count of

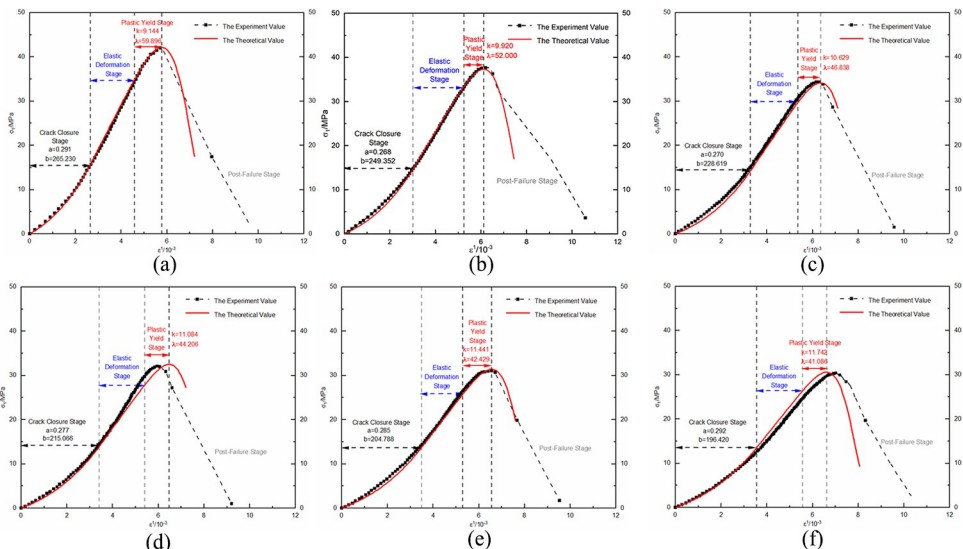

**Fig 13. Comparison between uniaxial re-loading test curve and damage evolution model curves for unloaded damaged samples under dry-wet and freeze-thaw cycling conditions:** (a) Initial sate; (b) After 1 cycle; (c) After 3 cycles; (d) After 5 cycles; (e) After 7 cycles; (f) After 9 cycles.

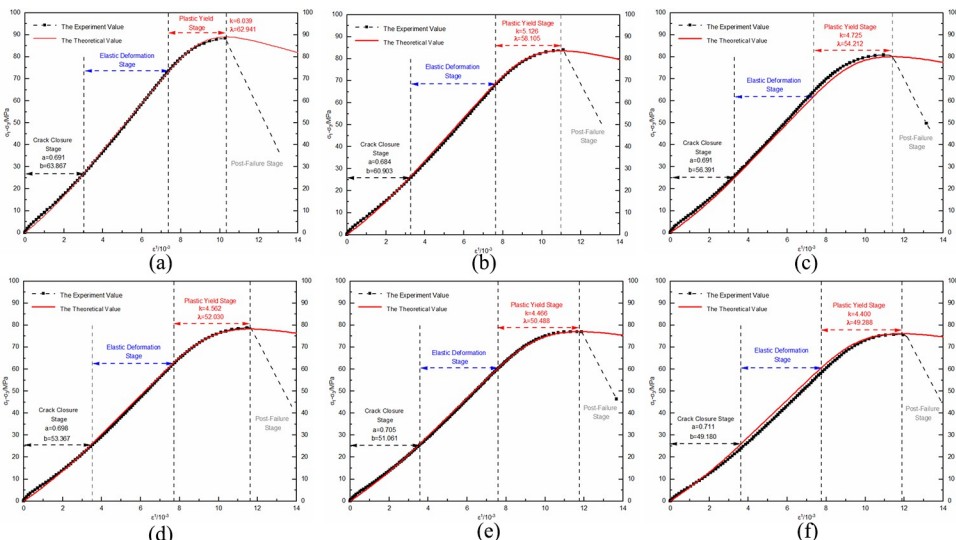

**Fig 14.** Comparison between triaxial re-loading test curve and damage evolution model curves for unloaded damaged samples under dry-wet and freeze-thaw cycling conditions: (a) Initial sate; (b) After 1 cycle; (c) After 3 cycles; (d) After 5 cycles; (e) After 7 cycles; (f) After 9 cycles.

cycles. Notably, in the uniaxial re-loading tests, the peak strain observed in samples subjected to 3 and 5 cycles is lower compared to those with fewer cycles, deviating from the expected trend of increasing peak strain with cycle number. This discrepancy highlights a variance between theoretical predictions and experimental observations. Factors such as the number of dry-wet and freeze-thaw cycles, coupled with confining pressure, play a pivotal role in the deformation and failure of sandstone that has experienced unloading damage. Furthermore, the accumulation of these cycles accentuates the development of pore structures in sandstone specimens that are not subjected to load. This phenomenon results in diminished peak strength values and reduces the elastic modulus noticeably. On the contrary, elevating the confining pressure seems to enhance the elasticity during deformation, while simultaneously restraining damage progression observed in samples that are not under load. This enhancement contributes to the plasticity characteristics—of the samples, indicating a transition from brittle failure modes toward more ductile behaviors. The patterns induced by changes in the numbers of environmental cycles underscore both the increase in porosity, leading to reduced peak strengths as well and diminished signs of rigid constituents. These can be observed through modalities in dips recorded in measured elastic moduli. However, under high applied pressures that hinder damage formation, mitigation processes can occur, resulting in improved scenarios beyond the sample's usual capacities. This leads to lasting deformations without immediate fracturing, suggesting potential adaptations towards ductile pathways rather than brittle ones, aligning more closely with the mechanics governing ductility as previously mentioned. The samples have the ability to depict the stress-strain variations leading to failure during the re-loading phase of unloaded damaged samples subjected to different dry-wet and freeze-thaw cycle numbers. This can provide a theoretical reference for relevant engineering applications.

## Numerical implementation of model

A three-dimensional finite difference method-based numerical simulation software has been employed to replicate the stress-strain behavior of samples with no load that have undergone

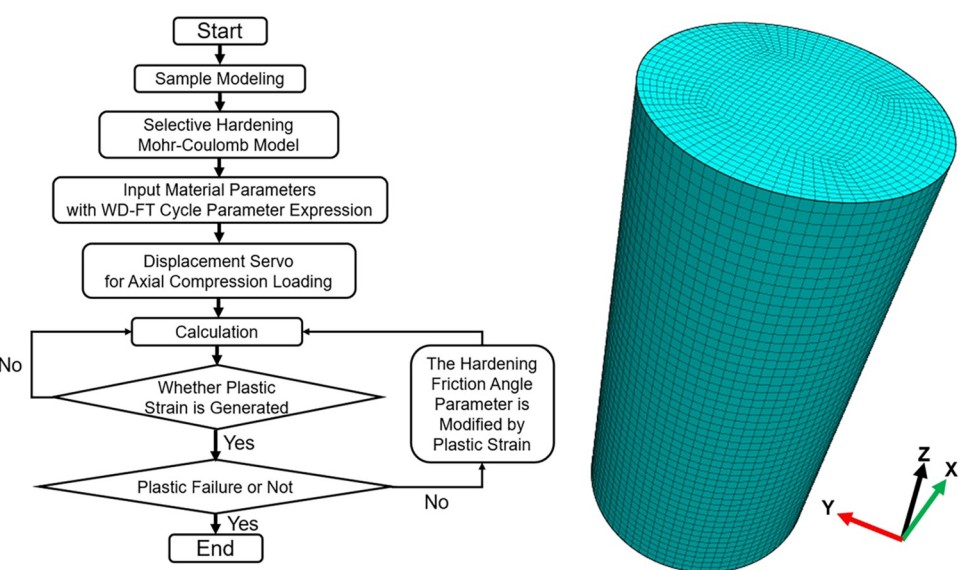

**Fig 15. Numerical model and calculation process for uniaxial compression test of sample.**

uniaxial compression testing following exposure to various dry-wet and freeze-thaw cycle conditions. The methodology for computation, along with the corresponding model, is illustrated in Fig 15. This model features a cylindrical geometry measuring 50 mm in diameter and 100 mm in height; it is subdivided into hexahedral mesh units each possessing a granularity of 2 mm across its structure which cumulatively contains approximately 60,000 elements. The constitutive model selected is the Hardening Mohr-Coulomb model. Axial loading is performed through velocity servo, and the stress increment and strain increment averages on the model's top surface are monitored. During the calculation process, each element is iteratively checked for plastic strain. Based on the current plastic strain of the element, the angle of friction is determined and the new hardened angle of friction is reassigned to the element, with the calculation terminating after plastic failure.

The enhancement behavior of the angle of friction in the plastic stage can be statistically predicted based on indoor tests. The corresponding angle of friction can be obtained by substituting the stress value at the plastic starting point into the limit equilibrium equation. The angle of friction at this moment is regarded as the initial non-hardened angle of friction, while the angle of friction determined at the peak point is considered the final hardened angle of friction. The statistical analysis of the percentage increase in the angle of friction corresponding to the plastic strain increment during the test is depicted in Fig 16. The percentage of enhanced angle of friction increases rapidly at the start of the plastic stage and then the rate of increase gradually slows down. The enhancement of the angle of friction during the plastic stage of single-axis re-loading of unloaded damaged samples under different dry-wet cycles remains within a certain range. Therefore, it is simplified and applied in the numerical simulation cycle for determining plastic strain.

## Numerical verification of model

Utilizing the data from uniaxial re-loading without prior cycling as an example, the comparison and analysis effects of the experimental curve in Fig 17, alongside the damage evolution theoretical curve and the numerical simulation curve. The numerical simulation phase does not reflect the pore compaction stage well, thereby the numerical simulation curve is adjusted

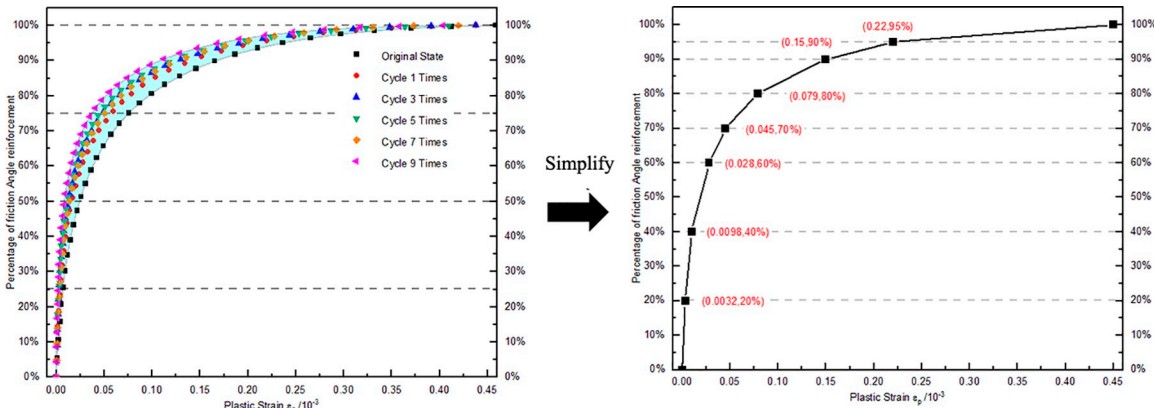

**Fig 16. Relationship between friction angle and plastic strain during the plastic yield stage.**

by adding the previously fitted pore compaction strain and shifting it to the right. Both the experimental and theoretical numerical simulation stress-strain curves agree well in the elastic and plastic stages, indicating that the fitting expressions obtained from the experiments are reasonably accurate. In Fig 18, different stress-strain curves for uniaxial re-loading of unloaded damaged samples under dry-wet and freeze-thaw cycling are displayed through numerical simulation, showing a nonlinear degradation trend at the peak points, reflecting the degradation process of dry-wet and freeze-thaw well.

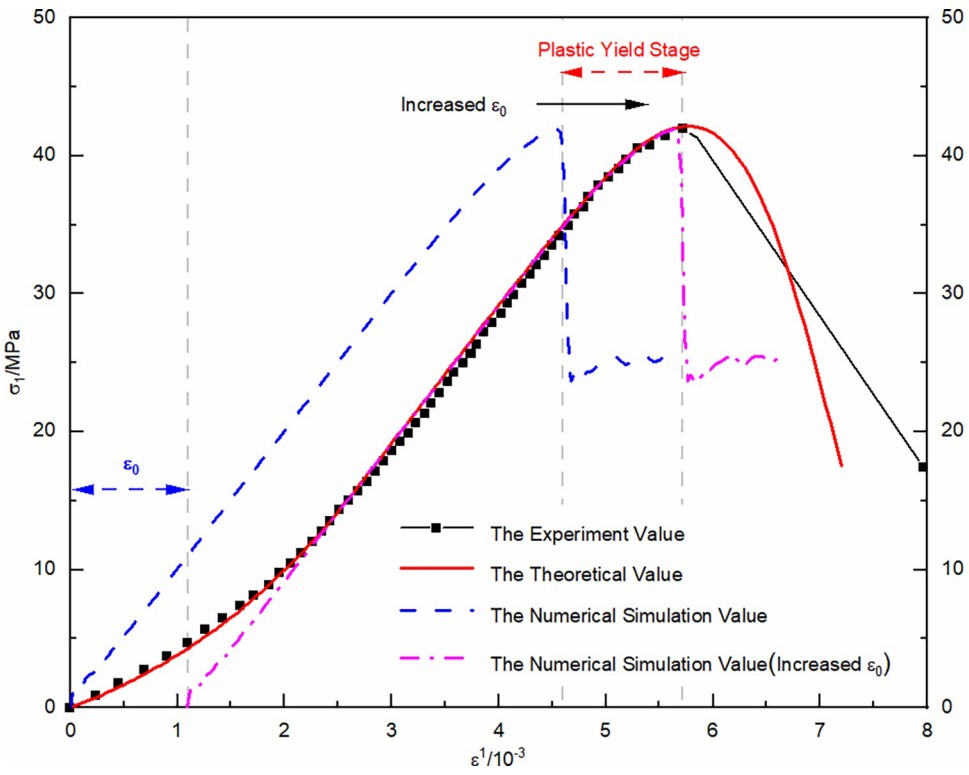

**Fig 17. Comparison of uniaxial re-loading stress-strain curve results for unloaded damaged samples under dry-wet and freeze-thaw cycling conditions.**

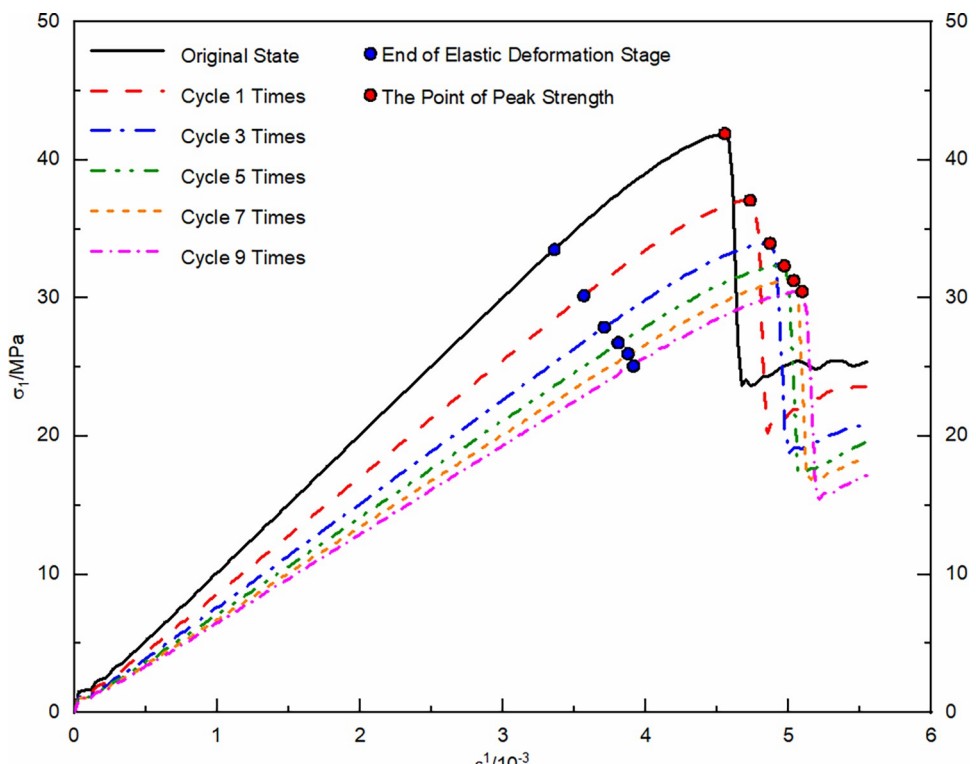

**Fig 18. Numerical simulation stress-strain curves for uniaxial re-loading of unloaded damaged samples under dry-wet and freeze-thaw cycling conditions.**

Failure mode analysis, guided by observations of the plastic yielding elements, was performed, with results in Fig 19. Observations reveal that with more frequent dry-wet and freeze-thaw cycles leading to compromised material strength parameters, there is a progressive rise in the count of plastic yielding elements concurrent with the failure of samples that have not been loaded. Simultaneously, the number of continuous plastic yield planes also increases. Typically, the failure surfaces manifest as conjugate or oblique shear modes, and with more

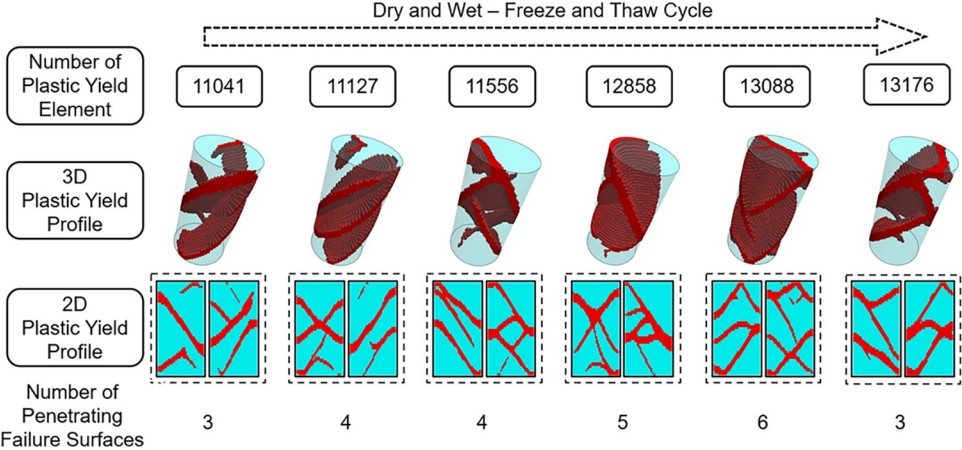

**Fig 19. Analysis of failure characteristics in uniaxial re-loading numerical simulation.**

such environmental cycles, the materials become increasingly fragmented due to expanding through-going plastic yield planes. Notably, the observed failure patterns bear resemblance to those recorded in experiments. When the combination of rupture surfaces cuts against the specimen surface, it can be assumed that the range is a potential region for the occurrence of dropout, which can be used to simulate the feed-flexing spalling damage occurring in the indoor tests. Although the numerical simulation and the actual can not be completely matched, the size of the dropped area increases with the number of cycles, and this law is consistent with the indoor experimental law. By drawing parallels between stress-strain relationships and failure modes, a high degree of concordance between numerical simulations and laboratory test data is found, endorsing the reliability of the macroscopic mechanical parameter alterations ascribed to varying dry-wet and freeze-thaw cycle frequencies. This concordance underpins the numerical simulations' effectiveness in reproducing the essential mechanics of the analyzed material.

## Conclusion

This paper focuses on the artificially excavated slope fluctuation zone's unloaded damage rock mass in the reservoir area as the research subject. Through a combination of indoor mechanical experimentation, theoretical analyses, and corroborative numerical simulations, this research has explored the laws governing changes in mechanical properties of damaged sandstone when it is not under load but subjected to cycles of drying-wetting and freezing-thawing. The result can be summarized as follows:

1. The stress-strain curve morphology for sandstone, which has undergone reloading post-damage and has been exposed to cycles of drying-wetting and freezing-thawing, generally maintains a consistent pattern. With an increment in cycle count, a downward trend is observed across the global stress-strain responses. There is a noticeable decrease in tangent slope growth during compaction, reduction in elasticity phase slopes, stretching of the plasticity stage with peak stresses gradually lessening while axial strains rise. These highlights pronounced degradation impacts on mechanical properties pertaining to unloaded damaged samples due to cyclic exposure to drying-wetting and freezing-thawing conditions.

2. With an increase in the number of dry-wet and freeze-thaw cycles, the elastic modulus, cohesion, and angle of friction of the unloaded damaged sandstone gradually decrease, generally showing a trend of steep deterioration initially followed by a more gradual decrease. The degradation of the elastic modulus is most pronounced, while the degradation of the angle of friction is the weakest. Under the first cycle, the degradation of the elastic modulus is about 10% to 18%, cohesion degradation is about 9.5%, and the angle of friction degradation is about 0.2%. The characteristics of failure in uniaxial re-loading are primarily brittle, with buckling and spalling damage becoming more pronounced as the number of cycles increases. Triaxial re-loading failure exhibits certain plastic characteristics, with the trend of plastic failure becoming increasingly significant as the number of cycles increases.

3. A damage evolution model for sandstone that has been unloaded and subjected to dry-wet and freeze-thaw cycles has been developed, drawing upon the principles of the statistical damage model. The predicted curve aligns well with experimental results, and the model demonstrates good predictive performance under different confining pressure conditions, indicating strong applicability. The model has been validated using numerical simulation software, showing consistency between numerical calculations, experimental results, and theoretical analysis, with the developmental patterns of failure characteristics also showing certain similarities to the experimental results. This indicates that the statistical fitting

expressions derived from laboratory experiments are reasonably accurate, and the numerical simulation calculation process is rational.

4. The research results indicate that the dry-wet and freeze-thaw cycling intervention results in gradual and cumulative deterioration of unloaded damaged sandstone. With an increase in the number of cycles, the structure of the sample becomes more relaxed, leading to a decline in its strength. The decline in the mechanical properties of the excavated, unloaded rock body will inevitably result in heightened deformation and reduced load-bearing capacity of the reservoir bank slopes. Therefore, it is imperative to fully and rationally consider all sequential influencing factors of the reservoir bank slopes to design more reasonable support measures and disaster prevention, mitigation, and risk management strategies.

## Author Contributions

**Conceptualization:** Xingzhou Chen.

**Data curation:** Wenrui Wang, Zhenkun Su.

**Formal analysis:** Wenrui Wang, Sheng Gong.

**Funding acquisition:** Xingzhou Chen.

**Investigation:** Lili Chen, Zhenkun Su.

**Methodology:** Wenrui Wang, Xingzhou Chen, Lili Chen.

**Project administration:** Xingzhou Chen.

**Resources:** Lili Chen.

**Software:** Wenrui Wang, Sheng Gong.

**Supervision:** Xingzhou Chen.

**Validation:** Wenrui Wang, Lili Chen, Sheng Gong.

**Visualization:** Wenrui Wang.

**Writing – original draft:** Wenrui Wang.

**Writing – review & editing:** Lili Chen, Sheng Gong.

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
