## [Decision Letter · Decision Letter 0]

17 Jul 2024

PONE-D-24-01541Study on the mechanical properties of unloaded damage sandstone under dry-wet and freeze-thaw cycling conditionsPLOS ONE

Dear Dr. Chen,

Thank you for submitting your manuscript to PLOS ONE. After careful consideration, we feel that it has merit but does not fully meet PLOS ONE’s publication criteria as it currently stands. Therefore, we invite you to submit a revised version of the manuscript that addresses the points raised during the review process.

We look forward to receiving your revised manuscript.

Kind regards,

Hui Yao

Academic Editor

PLOS ONE

“This research was funded by THE NATIONAL NATURAL SCIENCE FOUNDATION OF CHINA, grant number U1965107. These supports are gratefully acknowledged.”

“This research was funded by THE NATIONAL NATURAL SCIENCE FOUNDATION OF CHINA, grant number U1965107.The funders had no role in study design, data collection and analysis, decision to publish, or preparation of the manuscript.”

5. We note that your Data Availability Statement is currently as follows: [All relevant data are within the manuscript and its Supporting Information files.]

Reviewers' comments:

Reviewer's Responses to Questions

**Comments to the Author**

1. Is the manuscript technically sound, and do the data support the conclusions?

Reviewer #1: Yes

Reviewer #2: Yes

2. Has the statistical analysis been performed appropriately and rigorously? 

Reviewer #1: Yes

Reviewer #2: Yes

3. Have the authors made all data underlying the findings in their manuscript fully available?

Reviewer #1: Yes

Reviewer #2: Yes

4. Is the manuscript presented in an intelligible fashion and written in standard English?

Reviewer #1: Yes

Reviewer #2: Yes

5. Review Comments to the Author

Reviewer #1: 1. it is better to add some more references.

2. The amount of unloading is used to quantify the extent of unloading damage. The formula representing the amount of unloading should be presented.

3. Mechanical performance testing conducted after cycling test should be outlined.

4. Has pore characterization of rock samples before and after cycling test been assessed?

5. The research objectives should be clearly stated at the end of the literature review section.

Reviewer #2: 1. The table format needs to be standardized, as the border styles of Table1 and Table2 and 3 are not consistent.

2. In Figs. 11 and 12, the specimens only show cracks, and the failure degree of the specimens under different cycle times is not obvious in the graphs, try to improve the images to increase the recognition, such as adding colors to the peeling areas.

3. In Fig. 19, the failure characteristics are derived by numerical simulation, is it possible to establish a connection between the 3D or 2D model image and the experimental part of the previous paper, in order to better illustrate that the results of numerical simulation are consistent with the experimental results.

4. In this paper, the effects of dry-wet, freeze-thaw cycles on unloaded damaged sandstone were considered, and three dry-wet and one freeze-thaw cycle were selected in the cycling test methods. However, in the subsequent experimental tests, it was tested on specimens that completed a set of cycling steps. In the further study, it's better to carry out the experimental steps of dry-wet and freeze-thaw cycles separately, and analyze the specimens under three different cycling states of dry-wet, freeze-thaw, and both dry-wet and freeze-thaw, in order to better derive the effects of different and combined effects of dry-wet and freeze-thaw on unloaded damaged sandstone, so as to better analyze the mechanical properties of unloaded damaged sandstone under different seasons.

5. Some of the English writing grammar needs to be improved.

6. PLOS authors have the option to publish the peer review history of their article (what does this mean?). If published, this will include your full peer review and any attached files.

Reviewer #1: No

Reviewer #2: No

---

## [Author Response · Author response to Decision Letter 0]

7 Aug 2024

Dear Editor and Reviewers:

On behalf of the co-authors, we are very grateful to you for giving us an opportunity to revise our manuscript. Thank you for your decisions and constructive comments on my manuscript entitled “Study on the mechanical properties of unloaded damaged sandstone under dry-wet and freeze-thaw cycling conditions” (manuscript ID: PONE-D-24-01541) for your decisions and constructive comments on my manuscript. We agree with the suggestions of both reviewers and have incorporated the suggested changes into the manuscript.

The main corrections in the paper and the responds to the reviewer’s comments are as flowing:

Reviewer#1:

1.Comment: It is better to add some more references.

Response: We sincerely thank you for your valuable comments. We have carefully reviewed the literature and have added references on slope unloading damage and environmental factors affecting slope stability in the introductory section of the revised manuscript. Meanwhile, in the section of Construction of the Damage Model, relevant literature on damage modelling has been added.

(A total of 8 relevant papers have been added: see papers 2-6, 23, 24, 27 and 28, which are located at lines 37-40 on page 2, line 71 on page 4 and line 306 on page 18, respectively.)

2.Comment: The amount of unloading is used to quantify the extent of unloading damage. The formula representing the amount of unloading should be presented.

Response: Your input is essential to help us better understand and interpret the definition of "unloading capacity". Therefore, we have added the formula for defining unloading capacity and the physical definitions of the items in the formula. Unloading capacity is the percentage of the ratio between the original perimeter pressure minus the existing perimeter pressure and the original perimeter pressure minus the unloading damage perimeter pressure. The unloading quantity is defined as follows:

U_s=(σ_3^0-σ_3^i)/(σ_3^0-σ_3^f )×100% (1)

In the definition of unloading, U_s denotes the amount of unloading, σ_3^0 represents the initial perimeter pressure value before unloading, σ_3^i the target perimeter pressure value for unloading, and σ_3^f the perimeter pressure value at the time of unloading damage.

(Equation 1 above has been added to the manuscript content and can be found on page 5, lines 105 to 108.)

3.Comment: Mechanical performance testing conducted after cycling test should be outlined.

Response: Thank you for your careful review, the omission of the experimental method on cycling followed by reloading of the mechanical test undermines the integrity of the experimental method section. We have therefore added a text description of the reloading test after the description of the cyclic test setup. The purpose of the reloading tests is to obtain the mechanical properties of the post-cycling unloaded damaged specimens, and they are the same as the conventional single triaxial procedure for rock, with the noteworthy exception that before the triaxial loading we have to return the specimen's environment to the pressurised environment of the unloaded damaged specimens.

(The addition of the textual description of the reloading test can be found on page 7 of the manuscript along with lines 144 to 154 on page 8, which is at the bottom of Figure 3.)

4.Comment: Has pore characterization of rock samples before and after cycling test been assessed?

Response: As you say, we also believe that evaluating the pore characteristics of unloaded damaged specimens before and after cyclic action helps us to understand the nature of cyclic action. In order to ensure that the conclusions from the tests we designed were reliable, we measured the porosity of the specimens during the cycling tests.

The method of measurement was as follows, in order to avoid the influence of saturation and drying during porosity determination on our original test plan, we measured the volume and mass of the specimens after each drying/saturation during the wet/dry cycling process, and we obtained the porosity of the specimens during the cycling process by calculation.

We found that the porosity increased during the cycling process, which indicates that the structure of the unloaded damaged specimens is becoming looser and looser, and that there is a link between this and the mechanical properties of the specimens.

However, I did not describe this part of our work in the manuscript for two main reasons. The first is that we were unable to accurately characterise the porosity of the specimens after freeze-thaw cycling. According to the methodology for porosity determination, the measurement of porosity after freeze-thaw causes the specimens to undergo an unintended drying and saturation, and we were concerned that this would affect our ability to quantitatively characterise the cycling-mechanical property relationship. Therefore, I think we may need to incorporate some non-destructive testing methods to study it in further research, such as the common longitudinal wave velocity and electron microscope scanning methods. The second point is that, considering that the porosity we determined can only reflect the trend of change but is not precise enough, we did not include this item in our article during the process of building the damage model and compiling the manuscript.

5.Comment: The research objectives should be clearly stated at the end of the literature review section.

Response: We agree with you, and therefore we have modified the last paragraph of the introductory section of the manuscript. Our two research objectives, to investigate the variation of mechanical properties of unloaded damage specimens under dry-wet-freeze-thaw cycling and to develop a damage model that can be used to reflect this change in mechanical properties, have been described in a more direct manner.

(A description of the revised additions can be found after line 74 on page 4 of the manuscript.)

Reviewer#2:

1.Comment: The table format needs to be standardized, as the border styles of Table1 and Table2 and 3 are not consistent.

Response: We were really sorry for our careless mistakes. Thank you for pointing this out. We have standardised the format of the three tables.

(The revised tables are at paragraphs 11, 14 and 22 respectively.)

2.Comment: In Figs. 11 and 12, the specimens only show cracks, and the failure degree of the specimens under different cycle times is not obvious in the graphs, try to improve the images to increase the recognition, such as adding colors to the peeling areas.

Response: Your suggestions are very useful and important, which help to make Figures 11 and 12 more clear and concise, as well as to help us in interpreting the descriptions of the damage features. Thank you for your careful review. In Figure 11, we have used red stripes ( ) to mark the areas of flexural spalling damage in the sketch of the rupture features, and in Figure 12, we have used blue arrows ( ) to point out the location of the entry points of the diagonal damage. These modifications make it very clear that as the number of wet-dry-freeze-thaw cycles increases, the unloaded damage specimens show an increase in the area of flexural spalling damage and an increase in the number of crack incision points at the time of rupture.

(The revised diagram can be found on page 17.)

3.Comment: In Fig. 19, the failure characteristics are derived by numerical simulation, is it possible to establish a connection between the 3D or 2D model image and the experimental part of the previous paper, in order to better illustrate that the results of numerical simulation are consistent with the experimental results.

Response: This suggestion is very constructive, and indeed we are equally concerned about the link between numerical modelling and in-house testing. However, it should be noted that due to the limitations of the numerical simulation method (Finite Element Method), the results of numerical simulation and indoor experimental results can not be completely matched, but still can reflect a certain pattern of change. Although the numerical simulation can not simulate the collapse flexure spalling damage, but we found that the combination of the rupture surface of the specimen and the specimen surface as the specimen potential drop area, this part of the potential area with the increasing number of cyclic action and more broken, the area of the volume also increased. Therefore, we attempted to describe this phenomenon, but constructing a more explicit and quantifiable relationship still requires further research, which will be the goal of our further studies in the future.

(The addition of a section on numerical and experimental linkages can be found on pages 27 and 28, line numbers 430 to 434.)

4.Comment: In this paper, the effects of dry-wet, freeze-thaw cycles on unloaded damaged sandstone were considered, and three dry-wet and one freeze-thaw cycle were selected in the cycling test methods. However, in the subsequent experimental tests, it was tested on specimens that completed a set of cycling steps. In the further study, it's better to carry out the experimental steps of dry-wet and freeze-thaw cycles separately, and analyze the specimens under three different cycling states of dry-wet, freeze-thaw, and both dry-wet and freeze-thaw, in order to better derive the effects of different and combined effects of dry-wet and freeze-thaw on unloaded damaged sandstone, so as to better analyze the mechanical properties of unloaded damaged sandstone under different seasons.

Response: Thank you very much for guiding us in the direction of future research. Separating the dry and wet factors, the freeze-thaw factor and then quantitatively describing them, and secondly trying different combinations on the effect of unloading damage specimens will be our further research direction.

5.Comment: Some of the English writing grammar needs to be improved.

Response: We tried our best to improve the manuscript and made some changes to the manuscript. These changes will not influence the content and framework of the paper. And here we did not list the changes but marked in red in the revised paper. We appreciate for Reviewers’warm work earnestly and hope that the correction will meet with approval.

We sincerely appreciate the time and effort invested by the reviewers in evaluating our manuscript. We look forward to any additional feedback or suggestions.

Yours sincerely

Wenrui Wang

5-Aug, 2024

---

## [Editor Report · Decision Letter 1]

23 Aug 2024

Study on the mechanical properties of unloaded damage sandstone under dry-wet and freeze-thaw cycling conditions

PONE-D-24-01541R1

Dear Dr. Chen,

We’re pleased to inform you that your manuscript has been judged scientifically suitable for publication and will be formally accepted for publication once it meets all outstanding technical requirements.

Kind regards,

Hui Yao

Academic Editor

PLOS ONE
---

## [Editor Report · Acceptance letter]

1 Sep 2024

PONE-D-24-01541R1 

PLOS ONE

Dear Dr. Chen, 

I'm pleased to inform you that your manuscript has been deemed suitable for publication in PLOS ONE. Congratulations! Your manuscript is now being handed over to our production team.

Kind regards, 

on behalf of

Dr. Hui Yao 

Academic Editor

PLOS ONE